# Learning Corresponded Rationales for Text Matching

## Abstract

The ability to predict matches between two sources of text has a number of applications including natural language inference (NLI) and question answering (QA). While flexible neural models have become effective tools in solving these tasks, they are rarely transparent in terms of the mechanism that mediates the prediction. In this paper, we propose a self-explaining architecture where the model is forced to highlight, in a dependent manner, how spans of one side of the input match corresponding segments of the other side in order to arrive at the overall decision. The text spans are regularized to be coherent and concise, and their correspondence is captured explicitly. The text spans – rationales – are learned entirely as latent mechanisms, guided only by the distal supervision from the end-to-end task. We evaluate our model on both NLI and QA using three publicly available datasets. Experimental results demonstrate quantitatively and qualitatively that our method delivers interpretable justification of the prediction without sacrificing state-of-the-art performance. Our code and data split will be publicly available.

## 1 Introduction

Text matching serves as a key subroutine facilitating many natural language processing (NLP) tasks including natural language inference (NLI) (Bowman et al., 2015; Wang & Jiang, 2016a; Khot et al., 2018), paraphrase detection (Wang et al., 2017c), question answering (QA) (Andreas et al., 2016a; Wang et al., 2017b; Chen et al., 2017), and others. Much of the progress across such tasks has come from the use of flexible neural architectures that can be trained to achieve high performance. However, absent attention or rationale mechanisms, the resulting models lack transparency about how the decisions are reached. Soft attention keeps the model differentiable, typically operates on a word-by-word basis (Parikh et al., 2016; Seo et al., 2016; Chen et al., 2016b; Wang et al., 2016; Wang & Jiang, 2016b), but does not strictly confer importance as weakly attended words can impact decisions. In contrast, rationales (Lei et al., 2016) as hard selections are challenging to train but provide a certificate of exclusion of any unattended part. We opt for rationales in this paper as our goal is to identify clearly necessary text spans as opposed to individual words, and to do so across the matched texts in a dependent way.

Our work builds on and relates to many recent advances in explaining neural predictions for NLP (Lei et al., 2016; Li et al., 2016; Sharp et al., 2017; Koh & Liang, 2017; Alvarez-Melis & Jaakkola, 2017; 2018) but aims for rationalizing textual matching. Consider answer reranking or selection in the state-of-the-art open domain QA pipeline[1] (Wang et al., 2017b). The end goal in this case is to classify whether the supporting passage indeed has the answer to the specific question. Figure 1 illustrates the problem with two answer candidates and their supporting passages. A rationale that explains the correct match should not only highlight "*Galileo Galilei*" in the passage but also include two additional facts from the question and their corresponding evidence in the passage (highlighted in red and blue). In this paper, we seek rationales that identify necessary textual spans from each question and, in a dependent way, their short and sufficient corresponding segments in the supporting

---

[1]The multi-step QA pipeline that follows the process of search, read and re-rank. In brief, an information retrieval model first coarsely selects relevant passages to a given question. Then, a reading comprehension model infers a list of candidate answers from the passages, followed by a re-ranking algorithm that reorders the answer candidates based on aggregated evidence from different passages towards the final prediction.

**Question:** Who is the physicist , mathematician and astronomer that discovered the first four moons of Jupiter ?

---

**Answer candidate:** *Galileo Galilei*                                                    **Label:** positive
**Passages:** *Galileo Galilei* was an Italian physicist , mathematician , astronomer , and philosopher who played a major role in the Scientific Revolution . *Galileo Galilei* is credited with discovering the first four moons of Jupiter .

---

**Answer candidate:** *Isaac Newton*                                                       **Label:** negative
**Passages:** Sir *Isaac Newton* was an English mathematician , astronomer , and physicist who is widely recognized as one of the most influential scientists ...

---

Figure 1: Examples of the corresponded rationales for QA. Words highlighted in the same color from the question and a passage is a paired rationale. "*Galileo Galilei*" is the correct answer since two facts from the question are matched in its passage. On the other hand, the passage of "*Isaac Newton*" does not cover the second fact, which makes it a wrong answer.

passages. Only the selected text spans in aggregate – the rationales – are used as the basis for the final decision about matching.

We do not assume that any rationale annotations are given. Instead, the rationales are learned as latent selections based on the distal supervision (Mintz et al., 2009) from the downstream text matching task. The rationale mechanism builds on the generator-predictor framework Lei et al. (2016) introduced for sentiment analysis where the generator selects text spans as rationales that are then fed to the predictor. In our case, the generator is tasked with a combinatorial selection across the texts being matched. To make the combinatorial selection easier to learn we first identify key pieces from a single side of the input (*e.g.* question or hypothesis) and then look for the corresponding spans from the other side in a dependent manner. The predictor finally maps the extracted rationales in aggregate to task-specific values. The entire model along with the rationales is trained in an end-to-end fashion through policy gradient (Williams, 1992).

We consider both NLI and QA as the evaluation tasks. Extensive experimental results on three publicly available datasets show that the model delivers human-interpretable information while preserving comparable performance to the state-of-the-art. Further, to quantify the quality of generated rationales, we contrast the rationales arising from clean vs adversarial inputs. The generated rationales (as they operated over spans) are fairly robust against inserting irrelevant texts.

## 2 RELATED WORK

**Learning interpretable models**   Improving interpretability has been approached primarily from two different angles: learning self-explainable models and post-hoc prediction analysis. The former encodes explainability in the architecture itself, for example, by generating understandable intermediate results, while the latter aims to explain the mechanism of prediction in already learned models.

An example of learning with self-explainable architecture is the neural module network, introduced in Andreas et al. (2016b), and later extended Andreas et al. (2016a), Johnson et al. (2017), and Mascharka et al. (2018). The interpretability comes from the mechanism of composing appropriate modules following the logical program produced by a natural language component. The restriction to a small set of pre-defined programs currently limits their applicability.

Input-level selection is another popular way to generate human interpretable explanations. These approaches often integrate explanation generation as part of the learning problem. Lei et al. (2016) highlighted fragments of the input text as a rationale. In contrasts, Li et al. (2016) finds the minimal set of words that need to be erased before the decision changes. Our model is inspired by (Lei et al., 2016) but extends the ideas towards combinatorial corresponding rationales across matched texts. In other words, our rationales are structured, over a pair of inputs. Sharp et al. (2017) and Choi et al. (2017) also address interpretability of QA but focus on document selections instead of reasoning about the matching problem at the finer level. Ni et al. (2018) learns how to transform a natural-language question into a query for an IR model by identifying essential terms. But they assume that ground-truth selections are provided.

Efforts in learning post-hoc explanations include analyzing and visualizing neural activations (Hermans & Schrauwen, 2013; Li et al., 2015; Karpathy et al., 2015; Bau et al., 2017; Selvaraju et al., 2017), converting word embeddings into sparse or binary interpretable vectors (Faruqui et al., 2015), and approximating the model via locally interpretable forms (Ribeiro et al., 2016; Alvarez-Melis & Jaakkola, 2017; Lundberg & Lee, 2017; Lee et al., 2018). Recently, Chen et al. (2018) trained an instance selection model based on the class distributions from the model to be explained, which could be considered as a model-agnostic variant of (Lei et al., 2016). The approaches here provide a posteriori explanations for previously trained models while we focus on learning explanations in the course of model training.

**Attention models** Attention-based models (Bahdanau et al., 2014; Xu et al., 2015; Luong et al., 2015; Rush et al., 2015) offer another means to elucidate the inner workings of neural models. These models have been successfully applied to many tasks of sequence comparison. Parikh et al. (2016) proposed word-by-word attention model for NLI and later extended by Seo et al. (2016) for QA. Similar ideas also appeared in the work of (Chen et al., 2016b; Wang et al., 2016; 2017c). Kim et al. (2017) proposed a structured attention mechanism, which induces a latent tree structure and computes the alignment jointly. Later, based on the idea of comparing two sentences using their tree structures (Chen et al., 2016a; Zhao et al., 2017), Liu et al. (2018) extended structured attention for matching sentences.

The methods discussed here are primarily based on soft-attention instead of hard-selection. Although such attention is easier to train and softly simulates the alignment, it serves as a proxy in understanding how portions of the input contribute to final prediction. Absent additional regularization, any small attention score can still in principle contribute significantly to later computations through re-amplification in later stages. For the purpose of understanding neural predictions we opt for hard-selections despite the more challenging training (Deng et al., 2018).

# 3 RATIONALIZING TEXT MATCHING

We formalize here the problem of learning corresponding rationales. Consider a pair of word sequences, namely $q = \{q_1, \cdots, q_m\}$ and $p = \{p_1, \cdots, p_n\}$, where $q_i, p_j \in \mathbb{R}^d$ denote vector representations. The learning problem is to output a task-specific prediction $l$ based on matching $q$ and $p$. For example, in NLI, the task is to classify the hypothesis-premise pair as entailment, contradiction or neutral. On the other hand, the prediction task for QA could be in the form of answer generation, classification or ranking.

Our goal is to generate human-interpretable information to gain insight into the mechanism of prediction. Specifically, we aim to select a list of pairwise text-spans as corresponding rationales, namely $\mathcal{R} = \{(x^k, y^k)\}$, where $x^k$ and $y^k$ are the $k^{\text{th}}$ rationale pair selected from sequence $q$ and $p$, respectively. $x^k = \{x_1^k, \cdots, x_m^k\} \in \{0, 1\}^m$ and $y^k = \{y_1^k, \cdots, y_n^k\} \in \{0, 1\}^n$ are binary indicator sequences of lengths $m$ and $n$ where each nonzero entry denotes a selected word. The cardinality of the set $\mathcal{R}$ varies with the input pair. In order for these pairwise text-spans to qualify as rationales, we ask them to satisfy the following desiderata.

- Coherent and concise: spans selected from each side of $q$ and $p$ need to be short and coherent compared to the original text.
- Correspondent: for every pair of text spans, the selected words in $x^k$ and $y^k$ should be relevant and correspond to each other in a task-specific manner toward the end decision.
- Sufficient: the selected words in the rationale collection $\mathcal{R}$ should serve as a substitute of the full original texts for the predictive task without degrading performance

For example, in QA, each $x^k$ indicates a key factor that a particular question is asking, while $y^k$ is the corresponding supporting evidence in the passage. We call the selection of words as the rationale generator.

In this work, we assume that no rationale annotations are given. Thus, the rationale generation is learned entirely in an unsupervised manner, guided only by the distal supervision from the end task, in accordance with the above desiderata. The rationales are latent variables where selections from $q$ and $p$ are the combinatorial objects. The supervision from text matching guides the model to indirectly learn to choose the word combinations that provide good explanations for predictions that

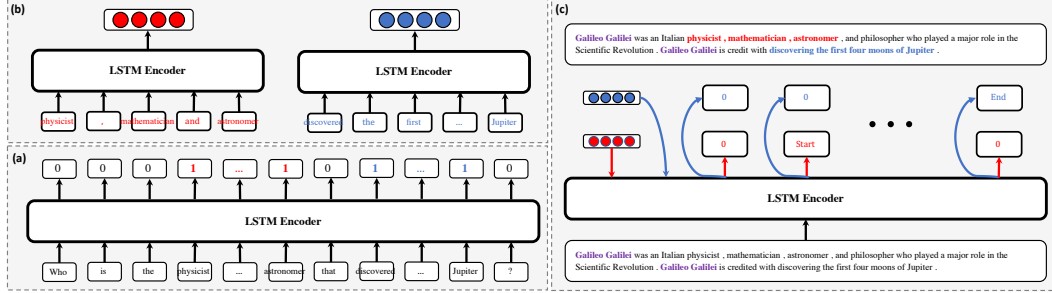

Figure 2: The architecture of the rationale generator. The model is comprised of (a) a module for identifying rationales on $q$, (b) a rationale re-encoding module, and (c) a component for generating corresponding rationales on $p$. The sentence encoder is shared across all modules.

are themselves learned concurrently. We will next describe the generator and predictor in greater detail.

## 3.1 RATIONALE GENERATOR

Selecting words from $q$ and $p$ with desired properties is a combinatorial problem. It would be infeasible to consider all possible selections across the sequences jointly. Instead, we propose a dependent selection mechanism that follows a generation-encoding-generation scheme. Concretely, we first identify all rationales $x^k$ from sequence $q$. For every $x^k$ we re-encode the selected words to construct a standalone rationale vector. These vectors are then used to find corresponding rationales from $p$. Figure 2 illustrates the architecture of our rationale generator.

**Identifying rationales from $q$** We aim to obtain a set of rationales $\{x^k\}$ from $q$. Let $z = \{z_1, \cdots, z_m\} \in \{0,1\}^m$ be the super-set of highlighted words from all $x^k$. We generate $\{x^k\}$ by first generating $z$, followed by partitioning. The problem of generating $z$ is cast as sequence tagging, where $z \sim p(z|q)$. In a simple case, we parameterize the probability distribution with a bi-directional long short-term memory (LSTM) network (Hochreiter & Schmidhuber, 1997). Each $z_i$ is selected conditionally independently given $q$. Thus joint probability factors according to

$$p(z|q) = \prod_{i=1}^{m} p(z_i|q) = \prod_{i=1}^{m} \sigma(W_z h_i^q + b_z), \tag{1}$$

where $h_i^q$ is the $i^{\text{th}}$ hidden state of the LSTM taking input $q$, and $\sigma(\cdot)$ denotes the sigmoid function. Once $z$ is selected, we partition the sequence by grouping successive ones to yield a list of new binary sequences $\{x^k\}$, all of the same length. The newly generated sequences satisfy the following two properties: 1) each $x^k$ contains only localized group of ones without intervening zeros; and 2) $z = \sum_k x^k$. For example, the partition outcome for the sequence "00111011" will be $\{00111000, 00000011\}$. It is worth mentioning that by construction, there is no textual overlap between different rationale selections from $q$, and that the number of rationales varies for different inputs.

**Rationale re-encoding** In order to find the corresponding rationale $y^k$ for each $x^k$, we re-encode the selected words of each rationale to a vector representation. This is done by feeding the element-wise product of $x^k$ and $q$ to the previously used LSTM. The LSTM's hidden states $h^{q*x^k} = \{h_1^{q*x^k}, \cdots, h_m^{q*x^k}\}$ are further masked by $x^k$ followed by max pooling to obtain the rationale vector. We denote the re-encoded rationale as $r^k = \max(h^{q*x^k} * x^k)$, and $*$ is the element-wise product. One may argue whether it is necessary to re-encode, $i.e.$, feed $q * x^k$ to the LSTM to get new hidden states so as to calculate $r^k$. An alternative way would be to directly use the hidden states from the LSTM of $q$ to obtain $\max(h^q * x^k)$, which requires less effort. However, even though the hidden states would be masked by $x^k$, we couldn't prevent information from other unselected words influencing $r^k$. This is due to the nature of bi-directional LSTM, where any hidden state might potentially (and often does) encode information beyond the immediate context. A similar argument would hold if we used a convolutional encoder assuming the length of consecutive rationale words were smaller than the width of the receptive field. Thus, re-encoding step ensures that the rationale vectors only contain information from the actually selected words.

**Generating corresponding rationales from $p$**  For each rationale vector $r^k$ generated from q, our goal is to find the corresponding text-spans in p. We encode the input sequence $p$ using the same sentence encoder, where the hidden states are now $h^p = \{h_1^p, \cdots, h_n^p\}$. To generate rationales, we could either use a tagging model as before or a boundary model (Vinyals et al., 2015; Wang & Jiang, 2016b) which marks the start and end tokens. Here, we introduce and use the boundary model, where the words in between the two tokens are used as the rationale. The probability of selection is simply

$$p(y^k \mid h^p, r^k) = p(y_{\text{start}}^k \mid h^p, r^k) \cdot p(y_{\text{end}}^k \mid h^p, r^k, y_{\text{start}}^k). \tag{2}$$

We produce the start and end tokens by first generating two head vectors

$$r_{\text{start}}^k = \text{ReLU}\,(W_{\text{start}} r^k + b_{\text{start}}), \ \text{ and } \ r_{\text{end}}^k = \text{ReLU}\,(W_{\text{end}} r^k + b_{\text{end}}). \tag{3}$$

followed by

$$y_{\text{start}}^k \sim p(y_{\text{start}}^k = i \mid h^p, r^k) = \frac{\exp\,(\langle h_i^p, \ r_{\text{start}}^k \rangle)}{\sum_j \exp\,(\langle h_j^p, \ r_{\text{start}}^k \rangle)}, \tag{4}$$

where $\langle \cdot, \cdot \rangle$ denotes the inner product. We sample the end token $y_{\text{end}}^k$ similarly, but conditioned on the selection of $y_{\text{start}}^k$ so as to ensure that the end selection appears only at or after the location of $y_{\text{start}}^k$. Compared to a tagging model, the rationale selected by a boundary model is by construction consecutive. In addition, the boundary model has the flexibility of not choosing anything if $p$ does not contain information corresponding to $r^k$. This can be achieved by selecting both start and end tokens at the "EOS" symbol in $p$.

## 3.2 Model Predictor

For simplicity, let us consider classification as the guiding task to instantiate the predictor, where both NLI and answer selection fall within the setup. The framework itself generalizes to other text-matching tasks. Given the list of generated rationale pairs $\mathcal{R}$ referring to input texts $p$ and $q$, our goal is to make these rationales alone useful for the final prediction. For each $k$, we first re-encode the rationale $y^k$ (as we did for $x^k$) and obtain LSTM hidden states $h^{p*y^k}$. We then construct a sparse matrix $S^k \in \mathbb{R}^{m \times n}$ as follows:

$$S^k = [(h^{q*x^k})^T h^{p*y^k}] * [x^k \otimes y^k], \tag{5}$$

where $\otimes$ is the outer product. The $(i, j)$ entry of matrix $S^k$ is nonzero if and only if words $q_i$ and $p_j$ are both selected in the corresponded rationale pair, *i.e.*, $x_i^k \neq 0$ and $y_j^k \neq 0$.

To predict the final label, we follow a strategy reminiscent of Match-LSTM (Wang & Jiang, 2016a). Concretely, for each $S^k$, we first normalize it using softmax in a row-by-row fashion to construct an attention matrix $A^k$. If $q_i$ is a rationale word in $x^k$, then $\sum_{j \text{ s.t., } y_j^k \neq 0} A_{ij}^k = 1$. Otherwise, the values in the corresponding row are all zeros. Next, we introduce aggregate representations for each $q_i$ as $\tilde{h}_i^{q*x^k} = \sum_j A_{ij}^k * h_i^{p*y^k}$. This can be interpreted as a measure of how $q_i$ can be matched by $p$. By construction, for each entry $i$ for which $x_i^k = 0$, $\tilde{h}_i^{q*x^k}$ is also zero. Following (Wang & Jiang, 2016a), we construct

$$h^k = [(h^{q*x^k} * x^k)\,; \tilde{h}^{q*x^k}\,; (h^{q*x^k} * x^k) - \tilde{h}^{q*x^k}\,; (h^{q*x^k} * x^k) * \tilde{h}^{q*x^k}], \tag{6}$$

where $[\cdot;\cdot]$ represents vector concatenation. We repeat the aforementioned construction for every $k$ and aggregate these representations to a unified vector $h = \sum_{k=1}^{|\mathcal{R}|} h^k$. The distribution over labels $l$ is then obtained by max-pooling on $h$ (per dimension) followed by a softmax linear prediction layer.

## 3.3 Joint Training

We minimize the discrepancy between $\hat{l}$ and $l$ via a cross-entropy loss $\mathcal{L}(p, q, \mathcal{R}, l)$, which makes generated rationales corresponded and sufficient. In addition, we want the generator to realize short and coherent rationales which is encouraged by two regularizers over selections $\{x^k\}$ and $\{y^k\}$ separately. For example

$$\Omega(\{x^k\}) = \frac{1}{m} \sum_{k=1}^{|\mathcal{R}|} (\lambda_1 \|x^k\|_1 + \lambda_2 \sum_{i=1}^m |x_i^k - x_{i-1}^k|), \tag{7}$$

where the first term penalizes the number of selected words, the second term makes the selections coherent and controls the total number of rationales generated from $q$. The regularizer is normalized by the length of the sequence. We introduce an analogous regularization for $\{y^k\}$. Note that the loss function depends on the sampled rationales in $\mathcal{R}$ and therefore we minimize the expected cost:

$$\min_{\theta} \sum_{(p,q,l)\in\mathcal{D}} \mathbb{E}_{\mathcal{R}} \left[ \mathcal{L}(p,q,\mathcal{R},l) + \Omega(\{x^k\}) + \Omega(\{y^k\}) \right], \qquad (8)$$

where $\theta$ is the set of parameters and $\mathcal{D}$ is the collection of training data. Directly minimizing the expected cost is hard due to discrete rationale selections and thus we appeal to policy gradient method (Williams, 1992) to derive stochastic gradients. Due to the loss and the regularization, the generated rationales in $\mathcal{R}$ are encouraged to satisfy all the three desired properties. It is worth mentioning that although the Gumble reparameterization trick (Jang et al., 2016) can alleviate training difficulties in similar settings, it won't offer the same certificate of exclusion of unselected parts. For this reason (clarity), we opt for keeping the hard selections also during training.

## 4 EXPERIMENTS

### 4.1 SETTINGS

**Datasets**  To evaluate the proposed method, we conduct experiments on three representative datasets. Specifically, we consider SciTail (Khot et al., 2018) as the benchmark for NLI and two QA datasets that are AskUbuntu (Dos Santos et al., 2015) and SearchQA (Dunn et al., 2017).

AskUbuntu is a non-factoid answer selection benchmark, which has been previously used for analyzing model interpretability (Lei et al., 2016). We follow the same setting and data split using both question body and title. On the other hands, SearchQA is an open-domain QA benchmark. The questions are Jeopardy-style queries, which are mostly clued sub-clauses and phrases separated by commas. Since the question is not in the format of natural language, we directly split the question with commas and take each segment as a question rationale. In other words, we assume the question rationale is given on SearchQA and the goal is to generate corresponded passage rationales. For simplicity, we only consider the answer selection problem in the multi-step open-domain QA pipeline (Wang et al., 2017b), where all candidate answers are generated by an existing reading comprehension model (Wang et al., 2017a).

Since SciTail provides data in form of both natural language and OpenIE tuple, we consider two setups: 1) we aim to generate corresponded rationales from the hypothesis-premise pair in natural language; 2) a similar setup as SearchQA that assume rationales on the hypothesis side is pre-defined base on the OpenIE results. We term these two settings as SciTail and SciTail$_{\text{factwise}}$, respectively. For SciTail$_{\text{factwise}}$, since the OpenIE extractions could be nested, we segment the provided tuple with the largest text coverage of the hypothesis and treat each argument as a separate rationale.

**Baselines**  We compare our approach to the following methods:

- Independent: The model treats two sequences $q$ and $p$ as independent texts. The generator first generates $\{x^k\}$ based on equation (1). And then, for each $x^k$, the generator generates $y^k$ regardless of the selection. After that, we use the same re-encoding procedure and the predictor.
- No rationalization: This method uses the entire text from both $q$ and $p$ without rationale generation, which reduces to the standard Match-LSTM algorithm (Wang & Jiang, 2016a).
- No rationalization w/ re-encoding: This method is only applicable to these experiments that the rationales of one side of text are given. Similar to "No rationalization", the method uses all the words as input. However, it re-encodes the rationales from $q$ and generates hidden representations based on each rationale assignments. Consequently, each piece of question/hypothesis rationale is encoded independently of the other pieces.

It worth mentioning that the last two baselines do not satisfy the purpose of learning corresponded rationales. However, their predictive accuracy serves as the upper bound of our proposed method.

**Implementation Details**  Unless specified otherwise, all the models are implemented with PyTorch (Paszke et al., 2017). The size of the hidden states of the LSTM encoder is 200 and the

| | SearchQA | | | SciTail$_{\text{factwise}}$ | |
| --- | --- | --- | --- | --- | --- |
| | % Alignment | Test MAP | Test Acc | % Alignment | Test Acc |
| Independent | $\sim 10\%$ | 58.3 | 49.6 | $\sim 20\%$ | 71.9 |
| | $\sim 40\%$ | 59.4 | 51.5 | $\sim 40\%$ | 75.1 |
| Ours | $\sim 10\%$ | 59.6 | 51.8 | $\sim 20\%$ | 74.2 |
| | $\sim 40\%$ | 60.2 | 52.9 | $\sim 40\%$ | 76.9 |
| No Rationalization w/ re-encoding | 100% | 59.9 | 52.3 | 100% | 78.0 |

Table 1: Experiment results on SearchQA and SciTail$_{\text{factwise}}$.

batch size is set to 40 pairs of texts. All models use fixed Glove 100-dimension word embeddings (Pennington et al., 2014), except for the AskUbuntu where we used the same word embeddings as (Lei et al., 2016). The maximum number of rationales we considered is three. For the case that the generator identifies more than three rationales on $q$, we extract the first three and discard the rest. We use Adam optimizer (Kingma & Ba, 2014) to train our model. We choose to use the tagging based model to generate rationales on $p$ except for the SciTail dataset. Other hyperparameters including the learning rate are tuned and selected according to the development set. We report the testing set results corresponding to the best development set settings. In addition, since both development and test sets of AskUbuntu are small ($\sim$200 questions), we report the averaged results over five runs.

Due to the training difficult of policy gradient, we also study an additional regularizer, which aims to make the rationale generator benefit from the soft-attention scores between words in $q$ and $p$. Specifically, we calculate a word-by-word similarity matrix using LSTM contexts and then normalized it in a row-by-row fashion. We denote the newly constructed attention matrix $A$. Then for each rationale $x^k$, we select the row of $A$ that $x_i^k = 1$. We rank all entries and pick the words in $p$ with top $K$ scores to form a new binary sequence $\hat{y}^k$ with length $n$. $K$ is set to be $\sum_i x_i^k$. We found that although $\hat{y}^k$ is insufficient to serve as rationales, it provides indirect supervision to help the model training. $\Lambda(\{y^k\}) = \lambda_3 \frac{1}{m} \sum_{k=1}^{|\mathcal{R}|} |y^k - \hat{y}^k|$ used as the additional regularizer for these experiments that the rationales on $q$ are provided. According to the development results, this regularizer helps SciTail$_{\text{factwise}}$ since it introduces contrastive information among words in $p$. But it fails to improve SearchQA. Thus, all the reported results on SciTail$_{\text{factwise}}$ include the additional regularizer.

## 4.2 MAIN RESULTS

We demonstrate that the proposed framework is able to match spans from on side of the input to corresponding segments by first reporting the performances of SearchQA and SciTail$_{\text{factwise}}$. We report the accuracy of both datasets. In addition, we also calculate the mean average precision (MAP) for SearchQA. We report performances at different highlight percentages on the alignment matrix, which is defined as $\sum_{k=1}^{\mathcal{R}} x^k \otimes y^k \in \mathbb{R}^{m \times n}$. With a proper re-encoding process, this quantity characterizes how much information is utilized for matching two sequences based on the corresponded information only. The sparsity control is achieved by constraining the percentage of selected words[2] in $p$ for each rationale generated in $q$.

Table 1 shows that our proposed method manages to select a small fragment of the alignment matrix while maintaining the accuracy. Furthermore, when selecting the same level of corresponded words, our dependent selection performs consistently better than the independent one. Surprisingly, our approach even outperforms the model without rationale generation on SearchQA when 40% of the alignment matrix is selected. This is might due to the fact that the passages are often redundant. Some examples of the generated rationales by the proposed model are shown in Figure 3. Given rationales in $q$, rationales in $p$ are selected as consecutive spans that with similar meanings. Such corresponded matches provide an interpretable way about how the model makes the prediction.

The next, we test the performance of our pipeline on AskUbuntu and SciTail and report results in Table 2. Since the average length of the question and passage for AskUbuntu is much larger, we are able to select much fewer words to predict the answer. Specifically, our dependent generation manages to use only 8% of the matchings to achieve 50.9 MAP. On the other hands, we select

---

[2]One way is to replace the $\ell_1$ norm in equation (7) to a hinge-loss that controls the percentage of selection to be less than a fixed level.

| | **AskUbuntu** | | | | **SciTail** | |
| | % Alignment | Test MAP | Test Acc | $|\mathcal{R}|$ | % Alignment | Test Acc |
|---|---|---|---|---|---|---|
| Independent | $\sim 8\%$ | 50.3 | 49.4 | 5 | $\sim 30\%$ | 74.9 |
| Ours | $\sim 2\%$ | 50.9 | 48.5 | 3 | $\sim 30\%$ | 76.0 |
| | $\sim 8\%$ | 51.5 | 50.0 | 5 | $\sim 30\%$ | 79.4 |
| No Rationalization | 100% | 53.5 | 51.7 | - | 100% | 82.2 |

Table 2: Experiment results on AskUbuntu and SciTail.

**Hypothesis:** Two light nuclei combine to produce a heavier nucleus and great energy in the nuclear fusion process
**Premise:** Nuclear fusion is when two or more light nuclei join together to form a heavier nucleus , releasing energy in the process

**Hypothesis:** protein is an important part of a healthy diet because it is needed to repair tissue
**Premise:** protein is needed in the diet for muscle and tissue growth and repair

Figure 3: Examples of the corresponded rationales extracted from the SciTail$_{\text{factwise}}$ dataset. Rationales are highlighted in different color. Text spans from the hypothesis and premise with the same color indicate a corresponded match. If a word in the premise is selected by two rationale pairs (*e.g.* red and yellow color), we change the text color (to red) and highlight (to yellow) accordingly.

30% for the shorter sentences in SciTail. The proposed rationalization approach outperforms the independent baseline, however, there is still a small gap compared to the Match-LSTM algorithm without any rationalization. In addition, for the SciTail dataset, we also investigate the choice of the maximum number of rationale we generate from $q$. This is denoted as the $|\mathcal{R}|$ in the table. We observe that by considering a larger number of rationale pairs, the overall performance gets improved. This because the larger number makes the sampling process of $z$ more robust.

## 4.3 Further Analyses

To analyze the quality of generated rationale in a quantitative way, we design an experiment that adds non-relevant sentences to the premise using the SciTail$_{\text{factwise}}$ dataset. Specifically, for each $(q, p)$ pair, we select a non-relevant premise sentence $p'$. We make sure that the corresponding hypothesis of $p'$ is not same to $q$. We construct a new premise by concatenating two sequences $[p; p']$, and then feed the new pair $(q, [p; p'])$ to a previously trained model for inference. Please note that the testing model does not see any adversary inputs with additive noise during its training. We hope the selected spans on the premise side should not cover any text in $p'$. To quantify the measure, for any rationale $x^k$ in $q$, we check if the corresponding rationale $y^k$ cover any words in the span of $p'$.

We conduct the aforementioned experiment on our pre-trained model. We first consider $(q, [p; p'])$ pairs that $q, p$ is a "entail" relation. We observe that among these inputs, our dependent rationalization model is fairly robust. Consider all selected words in $\{y^k\}$, there are 13.1% of words come from the irrelevant piece $p'$. Furthermore, in total, 13.0 % of the words in $p'$ are accidentally selected for at least once. These two numbers serve as a similar flavor of the "precision" and "recall" measures. Moreover, for these pairs that have "contradict" relations, we observe performance drops. Particularly, 16.7% of our selections are from $p'$ and 17.5% of the words in $p'$ are selected as rationales for at least once. The drop of performance might due to the difficulty of policy gradient optimization, which the selection model is hard to learn the option of not choosing any words for those rationales facts that are not covered by $p$. We will investigate the problem in our future work. Qualitative results of the above experiments can be found in Appendix A.

## 5 Conclusion

In this paper, we propose a novel self-explaining architecture to predict matches between two sequences of texts. Specifically, we introduce the notion of corresponded rationales and learn to extract them by the distal supervision from the downstream task. To evaluate, we compare our method to state-of-the-art on both NLI and QA. We show quantitatively and qualitatively that our algorithm delivers human interpretable justification while preserving high predictive performances.

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

# A EXAMPLES OF CORRESPONDED RATIONALES EXTRACTED FROM THE NOISY SCITAIL_FACTWISE TASK

---

**Hypothesis:** A ribosome consists of two elements , rrna and proteins
**Premise:** After mrna leaves the nucleus , it moves to a ribosome , which consists of rrna and proteins . *Thus , cell-free viral replication reactions are a proven technology which can be used for detailed biochemical and genetic dissection of the molecular mechanisms involved in the replication of picornaviruses*

---

**Hypothesis:** A hookworm is classified as a parasite
**Premise:** Hookworm is a parasite that lives in the gut and causes anemia . *Ecoregions are classified by biome type , which are the major global plant communities determined by rainfall and climate*

---

**Hypothesis:** Mollusks can be divided into seven classes
**Premise:** These materials can be divided into seven main groups . *Gases kinetic molecular theory : the theory that explains the behavior of ideal gases*

---

Figure 4: Examples of the corresponded rationales extracted from the noisy SciTail_factwise task. The *italic* sentences are the added noises. The first two examples are "entail" pairs while the last one is a "contradict".

