# OpenReview forum: "Learning Corresponded Rationales for Text Matching"
_ICLR.cc/2019/Conference_

### Official Review · AnonReviewer1 · 2018-10-30
**Interpretability is important; but without human data we cannot evaluate it**

**Rating:** 3
**Confidence:** 5

**Review:**

This paper proposes an approach to introduce interpretability in NLP tasks involving text matching. However, the evaluation is not evaluated using human input, thus it is not clear whether the model is indeed meeting this important goal. Furthermore, there is no direct comparison against related work on the same topic, so it is not possible to assess the contributions over the state of the art on the topic. In more detail:

- There are versions of attention mechanisms that are spare and differentiable. See here:
From Softmax to Sparsemax: A Sparse Model of Attention and Multi-Label Classification
André F. T. Martins, Ramón Fernandez Astudillo

- Why is "rationalizing textual matching" different than other approaches to explaining the predictions of a model? As far as I can tell, thresholding on existing attention would give the same output. I am not arguing that there is nothing different, but there should be a direct comparison, especially since eventually the method proposed is thresholded as well by limiting the number of highlights.

- A key assumption in the paper is that the method identifies rationales that humans would find useful as explanations. However there is no evaluation of this assumption. For a recent example of how such human evaluation could be done see:
D. Nguyen. Comparing automatic and human evaluation of local explanations for text classification. NAACL 2018
http://www.dongnguyen.nl/publications/nguyen-naacl2018.pdf

- I don't agree that explanations are sufficient if removing them doesn't degrade performance. While these two are related concepts, the quality of the explanation to a human is different to a system. In fact, more text can degrade performance when it is unrelated. See the experiments of this paper:
Adversarial Examples for Evaluating Reading Comprehension Systems.
Robin Jia and Percy Liang. EMNLP 2017: http://stanford.edu/~robinjia/pdf/emnlp2017-adversarial.pdf

- Reducing the selection of rationales to sequence tagging eventually done as classification is suboptimal compared to work on submodular optimization (cited in the introduction) if being concise is important. A comparison is needed.

- There is an argument that the training objective makes generated rationales corresponded and sufficient. This requires some evidence to support it.

- What is the "certificate of exclusion of unselected parts" that the proposed method has?

- An important argument is that the performance does not degrade. However there is no comparison against state of the art models to verfiy it.

---

> ### Author Response · Authors · 2018-11-29
> **Thank you very much for your valuable comments**
>
> A word-by-word soft attention is somewhat different, offering only an approximate version of the rationale we are after. We outline three reasons for this below.
>
> First, a soft attention does not provide any certificate of exclusion. By this we mean that any word receiving a small attention weight (as long as it is not zero) could be significantly amplified in later processing. We therefore cannot conclude that only words with large attention weights substantially influence the prediction.
>
> The second difference arises from how soft attention is typically computed. The attention is often based on context vectors associated with each word in q and p (e.g. hidden states of an LSTM). It is therefore unclear whether a particular attention score is driven by the surrounding context or the word in question. Put another way, if two specific text spans have high matching (attention) scores, this may no longer hold if we re-encoded those spans without their surrounding contexts.
>
> The third reason has to do with training. A thresholded word-by-word attention matrix, without re-training with the threshold, is neither sufficient (first reason) nor corresponded (second reason). Directly using the thresholded attention matrix as an explainer can therefore be expected to lead to a significant performance drop. Indeed, we specifically demonstrate this by evaluating a soft-attention model (MatchLSTM) and its thresholded version on SearchQA. The performance of these two models on the development set is 54.8 and 50.2, respectively. It is not clear how to best re-train a thresholded attention. In fact, it would seems somewhat challenging without resorting to exactly the type of routines we proposed in the paper.

---

### Official Review · AnonReviewer2 · 2018-11-03

**Rating:** 4
**Confidence:** 4

**Review:**


This paper tackles the problem of generating rationales for text matching problems (i.e., two pieces of text are given). The approach is in a similar spirit as (Lei et al, 2016) while the latter mainly focuses on one piece of text for text classification problems and this work focuses on generating pairs of rationales. The approach has been evaluated on NLI and QA datasets, which demonstrates that the generated rationales are sensible and comes at a cost of accuracy.

The approach employs a generation-encoding-generation schema: it firsts generates the rationale from one side as a sequence tagging problem, re-encodes the rationales and predicts the rationale on the other side as a span prediction problem. Leveraging a match-LSTM framework and generated rationales for prediction, the model can be trained using a policy gradient method.

Overall, I think this problem is novel and interesting. However, I am not fully convinced whether the proposed solution (and its implementation) is the right way to do so. Also, the paper writing needs to much improved.

First of all, there is certainly a drop in the end task performance while it is unclear whether the derived rationales are really that useful (if the goal is interpretability) in the current evaluation. I am not convinced by the noisy SciTrail evaluation for rationales -- the noisy part p’ can be totally irrelevant and assume that the rationale generation component learns some sort of alignment between two parts, so it is not surprising that the model will not select words from p’ and it doesn’t really show that the rationales are useful. I think it is necessary to conduct some human evaluation for generate rationales and also provide some simple baselines for comparison (for example, just converting the soft-attention in math-LSTM to some hard selections) and see if this interpretability (at a cost of task performance) is really worthy or not.

Secondly, I am not sure that whether the current way of generating the rationale pairs really makes sense or not.
It casts the rationale generation on one side as a tagging problem while the rationale generation on the other side as a span prediction problem. Why is that? Do you make any assumption that the two pieces of texts are not symmetric (e.g., one side is much longer than the other side like most of the current QA setup)?

There is a regularization term for both x and y but it seems that there isn’t any constraint that the generated rationales on the y side are not overlapping. Is it a problem or not? I don’t know how this is dealt with in the implementation.

Understanding sec 3 takes some efforts and I think the presentation could be much improved. For example, q * {x^k} is not defined -- I assume it means extracting the subset of q based on the 1’s in {x^k}. The equations in Sec 3.2 can be made clearer.

Finally, it is also unclear that how the 3 datasets were chosen. There are so many NLI and QA datasets (some of them are more popular and more competitive) at this point. Is there a reason that these datasets were chosen? There is a setup called ‘no rationalization w/ re-encoding’ which means that the rationale is already provided on one side, but is unclear that whether the OpenIE tuple and the searchQA queries can be used as rationales directly.

Minor points:
- Distal supervision -> distant supervision
- The first paragraph of Introduction, “absent attention or rationale mechanisms”, what does it mean by ‘absent attention’? Isn’t it the case that all the models used attention mechanisms?

---

> ### Author Response · Authors · 2018-11-29
> **Thank you very much for your valuable comments**
>
> Thank you very much for your valuable comments.  We believe there is some misunderstanding in terms of the difference between our rationalization model and attention-based model.  Please refer to the comments to Reviewer 1 for details.   Your other concerns are addressed below.
>
> 1. Adversarial evaluation on SciTail
>
> Intuitively, if the added noise p’ is totally irrelevant to the context of p, the predictive model should not rely on any information from p'.   However, it is very hard to achieve by many conventional models, especially for these based on word-by-word attention.  For example, consider the Match-LSTM that computes the contextual similarity between each word from both sides of the text.   And then, it constructs aggregated representations for the final prediction using soft-attentions normalized from the similarities.  Even the attention scores on p' are small, the aggregated representations cannot eliminate effects from these texts.    And the worse thing is that often the attention scores on p' are not very small, especially when the same word/entity appears in both q and p' (even the contexts are totally different).   This is one potential reason why many QA models are vulnerable to adversarial examples by appending noise, which is shown in the paper of "Adversarial Examples for Evaluating Reading Comprehension Systems" by Robin Jia and Percy Liang.   Of course, not selecting text from p' does not mean the selected rationales are correct.  However, it at least demonstrates the generated rationales from our model are fairly robust to adversarial noise.
>
>
> 2. Tagging v.s. span prediction
>
> Our rationale generation is formulated as a tagging problem on q and span prediction problem (use <start> and <end> tokens) on q.   The reason why we consider a tagging problem on generating rationales of q is that it can easily achieve the need of the variable number of rationales (via petition).  Then, given a rationale from q, we consider the corresponded rationale generation as span prediction is because that the generated rationale is guaranteed to be consecutive, which is much easy to optimize (via policy gradient) compare to sequence tagging with a continuity regularizer.
>
>
> 3.  Overlapping y
>
> There is no explicit regulation to enforce each rationale $y^k$ to be mutually exclusive text span.  The selection of y mainly depends on the selected x.  Different x might correspond to the same piece of text.  On the other hand, the sparsity loss is summed over all different $y^k$, which is implicitly to avoid overlapping (e.g. prevent the trivial case that all y^k selects the same text that is the entire p).
>
> 4. Dataset and OpenIE as the rationale
>
> As discussed in Section 4.1, we choose AskUbuntu because it is the only text matching dataset used in previous literature on (independent) rationale extraction. We choose the other two mainly because they could provide the "Factwise" settings and help better evaluate the "sub-task of generating corresponded rationales from p".
>
> We agree that it is difficult to tell whether the OpenIE tuples and SearchQA queries can be used as rationales from the linguistic view. However, first, empirically they provide us reasonable results when the "No Rationalization w/ re-encoding" was applied. Second and more importantly, for the "Factwise" experiments, i.e., the rationale is already provided on one side, we mainly hope to have a setting on only the sub-task of "generating corresponded rationales from p". In this way, we have a closer analysis of the performance and difficulties of this sub-task. This sub-task itself is not studied in previous work. We believe it is important to provide such results along with the end-to-end results.

---

### Official Review · AnonReviewer3 · 2018-11-06
**Interesting model with somewhat promising experiments, could benefit from some more comparisons**

**Rating:** 6
**Confidence:** 4

**Review:**

This paper is about learning paired rationales that include the corresponding relevant spans of the (question, passage) or (premise, hypothesis).  Experimental results show the same or better accuracies using just the fraction of the input selected as when the whole input is used.

While there has been prior work on learning rationales, this is the first I have seen that included this fine-grained pairing.  The paper also learns these rationales without explicitly labeled rationales but rather with only the distant supervision of the overall question answering or natural language inference task.

This paper could be made stronger by including an experimental evaluation of accuracy in an adversarial setting.  The model developed here might be well-suited for adversarial SquAD examples in which an extra sentence has been added.  It would be interesting to see these results.  This paper does include a somewhat similar adversarial evaluation (Section 4.3) but adds extra information to NLI examples.  Since for NLI, unlike QA, the extra sentence can change the correct label (can flip from entailment to contradiction), accuracy was not able to be evaluated.

Experimentally, it would be good to compare against some prior work that doesn't include the pairing.  Perhaps an interpretability model based on the passage only without fine-grained pairing with the question?  My apologies if this corresponds to "Independent", I was somewhat confused by descriptions of the baseline.

The descriptions of the baselines was the least clear part of this paper.  It would be helpful to improve the clarity of Section 4.1 (perhaps adding a figure).

Optional suggestion: consider breaking up the experiment section into two subsections: one for the cases in which the question rationales are provided (results in Table 1), and one for the cases in which the question-side rationales are learned as well (Table 2).  By putting all descriptions together, the paper explains two different settings and then needs to discuss which baselines are applicable to each setting and dataset and why.

---

> ### Author Response · Authors · 2018-11-29
> **Thank you very much for your valuable comments**
>
> Thank you very much for your valuable comments.
>
> In terms of the concern about the extra sentence can change the label of NLI, we agree that there could be the case.  But the chance is small since in our experiment we always select the extra premise, which pairs with a different hypothesis.
>
> We also agree that the setting like adversarial SQuAD could be potentially a better choice for the evaluation. However, this paper focuses on general text matching problem.  Although SQuAD relies a lot on text matching, it also requires additional engineering for QA.  For the future version, we will consider adding additional experiments on adversarial QA or other hypothesis verification task (like FEVER,  where a hypothesis holds as long as one passage can be found supporting the hypothesis. In this way, extra sentences won't change the label of the verification task).
>
> Moreover, the “independent” baseline extracts rationales on passages without access to fine-grained question rationale pieces.  The extraction model re-implements prior work of (Lei et al., 2016).   Intuitively, this method only learns to extract generally useful textual patterns for the task no matter what the questions/hypotheses are.  The extraction could be still useful for narrow domains like AskUbuntu, in which these patterns are limited. While in the open-domain setting and with the decrease of rationale sizes, its performance drops dramatically.  Thus extracting corresponded pairing rationales is crucial.

---

### Meta-Review · Area_Chair1 · 2018-12-14

**Confidence:** 4
**Recommendation:** Reject

**Metareview:**

This paper attempts at modeling text matching and also generating rationales.  The motivation of the paper is good.

However there is some shortcomings of the paper, e.g. there is very little comparison with prior work, no human evaluation at scale and also it seems that several prior models that use attention mechanism would generate similar rationales.  No characterization of the last aspect has been made here.  Hence, addressing these issues could make the paper better for future venues.

There is relative consensus between the reviewers that the paper could improve if the reviewers' concerns are addressed when it is submitted to future venues.